# Sense of Place, Biocultural Heritage, and Sustainable Knowledge and Practices in Three Italian Rural Regeneration Processes

Letizia Bindi [1,*], Mauro Conti [2] and Angelo Belliggiano [3]

1   Department of Social, Human and Learning Sciences, University of Molise, 86100 Campobasso, Italy
2   Department of Political and Social Sciences, Università degli Studi della Calabria, 87036 Cosenza, Italy; mauro.conti@unical.it
3   Department of Agricultural, Environmental and Food Sciences, University of Molise, 86100 Campobasso, Italy; belliggi@unimol.it
*   Correspondence: letizia.bindi@unimol.it

**Abstract:** This paper addresses sustainable development processes based on biocultural heritage, sense of place, and socio-cultural innovation and inclusiveness in the rural context, particularly focusing different scales of endogenous/neo-endogenous rural regeneration processes. Ethnographic and grounded case studies allow a critical analysis of different forms of rural development from a multidisciplinary perspective based on old and new rurality, rural and local communities' participation, resilience and regeneration processes, sense of place, belongingness, and "restanza". The three cases are situated in three areas of Italy: the ecomuseum of pastoralism in Pontebernardo in the northern region of Piedmont as a driver of local shared development pathways; the municipality of Castel del Giudice, in the Central-Southern region of Molise, enabling different and integrated local regeneration actions; and the Association "Casa delle AgriCulture"/Green Night Festival in Castiglione d'Otranto in the Southern region of Puglia as a performative opportunity to define innovative and transversal as well as community-oriented activities. In these contexts, different local regeneration programs were applied in order to provide a critical evaluation of the knowledge and practices of sustainability in terms of their entanglements with biocultural heritage and socio-cultural innovation. The present analysis insists on the impact of biocultural heritage in regeneration processes in rural regions and endogenous/neo-endogenous factors in rural sustainable development.

**Keywords:** bio-cultural heritage; sustainable development; local regeneration; participatory processes; new peasantries; social innovation; multifunctional rurality; ethnography; knowledge-practices system

## 1. Introduction

In this contribution we propose a first assessment of international research and training work experience that we carried out in the framework of an Erasmus + CBHE project, E.A.R.T.H. Education Agricultural Resources for Territories and Heritage, summarizing the monitored multi-situated contexts and the theoretical and methodological issues and aiming at a critical and innovative approach to the collected data.

First, we adopt a radically multi-disciplinary approach where the knowledge and methodologies of the social sciences, namely, Cultural Anthropology, Rural Sociology, Human Geography, and Communication Sciences, have been associated with those of economic geography and rural economy, animal husbandry, ecology, and law.

Second, we use an ethnographic methodology based on case studies, deep mapping, discourse analysis of the political frameworks, and symbolic capital activated in the various contexts observed.

Third, we attempt to model socio-economic behaviors and major trends in sustainable and unsustainable local development.

The final goal of this work is to elaborate a critical interpretation of three different development experiences and of the processes of involvement and participation of communities between success and failures, rethinking the idea of local development as a common good.

The analysis is therefore focused on different forms of endogenous and neo-endogenous territorial development, observing the causes and effectiveness of the collective actions and implemented local and supralocal policies and productive tourism models in different rural contexts as elements of the value chain of specific rural products and tourist destinations.

Through a multi-scale, multi-actor, and multi-situated analysis and modeling of cases, we tried to understand how:

(a) one case outlines a proactive ability to involve local communities in development processes;
(b) another seems essentially traced and hegemonized from above;
(c) in a third, outsiders/external actors play a crucial role in reversing local trends of economic depression and social isolation.

### 1.1. Social Sciences and the Analysis of Local Development Processes

Rural areas are presently at the center of local development policies as well as of socio-anthropological ethnography in which shared knowledge and local practices are addressed throughout deep mapping and participatory inventorying. Indeed, rurality and peasant cultures are themselves critical categories [1] in the framework of political anthropological studies from the 1950s and 1960s through the broadening 'from tribal to peasant' [2]. Even before [3], observing the peasant cultures as part of larger societies and across the different historical passages such as traditional, early modern, and early industrial agriculture, late modern, and "post-peasant" communities has always resulted in "dubious generalizations that cannot survive critical scrutiny. It is also a serious impediment to the formulation of theory in economic anthropology and microdevelopment" [4].

Trying to go beyond this notion of "peasantry" as a "mixed bag", reflection has developed over the decades across several specific ethnographies; however, this has rarely led to systematic comparison or general notions, because "the literature on peasant communities is not, at the moment, a very likely place to look for ideas, and certainly not for systems of ideas" [4], criticizing at the same time both the category of peasantry than the notion of modernity opposed, simplistically enough, to the one of backwardness that the anthropological research has opportunely discussed and criticized in the last decades.

Within the last twenty years, the social sciences in general and particularly public anthropology have focused around research actions on the local cultural heritage in its interconnections with the socio-economic development of regions even very peripheral to the predominant economic flows and sustainable tourism in rural areas as a form of diversification and multifunctionality of economic and productive activities [5].

Growing use has been made in the last decade of naturalistic images to speak of local community revitalization and development: a trend to a revival of biological and organic metaphors inscribed in an intellectual trajectory of re-naturalization and re-territorialization of identity dramatically impacting on the socio-economic dimensions of the targeted rural, inland and mountainous areas.

The first bursting metaphor was undoubtedly that of "resilience", a term taken from the physics of matter: "the ability of a material to absorb a shock without breaking," being subsequently absorbed and reworked, not by chance, from the ecology that uses it to indicate "the speed with which a biotic community is able to restore its stability if subjected to perturbations" and from psychology, especially from social psychology that speaks about" ability of an individual to face and overcome a traumatic event or a period of difficulty" [6].

The notion of 'resilience' has been questioned in the very last years as a plastic and dynamic form of adaptation and somehow as a strategy to "de-centering history" through "anthropogenic management" [7], as well as to escape to the hegemony of a historical approach to development processes.

Alongside this reworked and multidisciplinary notion of resilience, there is increasingly that of "regeneration" as the "the ability of many organisms to reconstruct or repair lost or damaged parts" [8], once again an image taken from plant biology.

Both these images are taken from the natural world and are translated, according to a late-positivistic narrative register into sometimes very conflictual processes: a very effective rhetorical artifice, though contemporarily quite critical. In fact, while natural self-repair, regeneration, and resilience are reactions independent of decisions and wills, the social phenomena of revitalization cannot be reduced to simplistic and univocal processes and notions; on the contrary, it must be understood as a complex form of reconciliation toward heritage-based forms of political capacity-building and environmental sustainability [9].

In this sense, the genesis of this radical reflection must be sought in certain important lines of interpretation and critical elaboration that we will try to briefly summarize.

One of the main theorical references is offered by Michel Foucault's crucial reflection on *Governamentality and Liberalism* [10] and on the *micropolitics of power* [11], aimed at grasping the subtle and embedded/embodied dynamics of influence, domination and conditioning of community choices by multi-scale institutional governance.

At a methodological scale, deep mapping and participatory inventories of intangible cultural heritage enable a rich data collection, the conservation and valorization of traditional knowledge on the environment and nature (Traditional Ecological Knowledge, or TEK). Bio-cultural notions, beliefs and behaviors are today reconsidered as local communities' cultural property contributing to more sustainable ways to inhabit, rational management of resources, and the subsistence of the communities themselves [12]. Bio-cultural heritage is a notion matured in the framework of different critical traditions: on one hand the latin american reflection on minority cultures and their knowledge–practices system as a form of embedded expert form of managing natural resources [13], and even before approached and discussed in studies upon biocultural diversity [14] as well as embodied/embedded perceptions of the environment and the way individuals manage and relating to them [15].

Counter-movements have arisen against the increasing commodification of intangible cultural goods, insisting "that cultural property should not be governed entirely by the market-based economic consideration" [16], as well as against the fact that TEKs have been, in the last two decades, "increasingly constructed and reified, romanticized and rhetorically exaggerated" [17–19].

Other relevant reflections have focused on alternative/new peasantries [20,21], on food networks and food citizenship [22], on rural innovation both on the level of environmental sustainability (agro-ecology, low-impact organic, bio-dynamic productions, etc.), and on social innovation (fair and responsible horizontal distribution networks, inclusion of migrants and other disadvantaged subjects in agricultural work and production activities, housing/inclusiveness in rural contexts). Thus, agriculture seems the ideal context for experimenting with hybridization processes between cultural and social traits, conservation of biodiversity, politics and governance of territories, and the agency and relationality of rural contexts [12].

This is evident in mountain/inner/fragile areas where multifunctional activities are planned aiming at connecting agri-food production, landscape conservation, ecosystemic services, new forms of tourist accommodation and housing, and artistic and entrepreneurial creativity.

Thus, resilient local communities have been opposed to the large extractive agro-industry, following a model close to, even if not immediately comparable, of indigenous and native peoples who were the bearers of antagonistic instances [23] against large "monocultures of the mind" [24] imposing a hegemonic, colonial, extractive and predatory form of rural exploitation.

The protection and enhancement of bio-cultural heritage is rooted, moreover, in the policies of recognition, connecting fragile and inner communities of the European rural space engaged in "political ecology of agriculture" [14] to post-colonial and de-colonial issues.

In this sense, levels of participation of rural communities become a powerful indicator in decision-making processes about local regeneration. The participatory paradigm has become, at least rhetorically and in the framework of European and transnational development cooperation calls, a re-assuring key notion systematically evoked in cooperation programs.

In many European and Italian regions community participation assures protection of local specificities and strengthens the sense of belonging of communities to the common history of the territory. Meanwhile, reputation and niche product valorization and branding coupled with territorial marketing and certification of origin appears, in turn, as a global heritage regime for food-oriented tourism. Thus, agriculture is a practice related to nature that at the same time deeply influences human communities in their use of natural resources and different forms of domestication of wilderness. One of the most interesting and recent approaches to the concept of protected areas and parks is that they are expressions of a specifically Western thought of compartmentalization and extraction of natural resources and of the resulting critical reflections on a "third landscape" [25–27].

Ecological discourse and the increasing narrative of the "green transition" are affected, in effect, by an urban-centered reformulation of the discourses on rurality and nostalgia, providing a symbolic "patina" of the past [28] to agricultural practices and non-industrial pastoralist societies that are thereby transformed into commodified and spectacularized heritage items ready to be symbolically used in the market of rural territories and tourist destinations. In effect, farmhouses, paths, restored country houses, and the experiential/slow/taste tourism market are increasingly based on environmental aesthetics and new styles of food consumption as well as on concepts of awareness, sovereignty, ethics, and citizenship.

## 1.2. A Critical Analysis of Local Development Policies

In this new framework, an analysis of the symbolic capital of agricultural production becomes particularly relevant; it requires the observation of new local subjectivities, institutionalized and/or informal protocols and rules of production, and transformation processes of raw materials as well as multifunctional rural activities.

This new outlook on rurality is established through multi-scale local policies, from the overall CAP frames to the proximity of the Local Action Groups (LAGs)' actions within the LEADER approach. In Italy, other levels of intervention have more recently been associated with them, specifically oriented to the areas of greatest fragility, such as the SNAI [29,30] and the CIS [31], as well as more recently all actions explicitly addressed by the National Strategy for Inner Areas (SNAI) and the Italian National Plan for Recovery and Resilience (PNRR). In each of these institutional programs, the emphasis is on overcoming the exogenous (or top-down) approach in favor of an endogenous one.

The LEADER approach is mostly based on the improvement of agricultural structures and on the strengthening of their competitiveness [32,33]. Several limits are recognized: strong dependence on public funding, a distorting effect generated by the privileges granted to certain sectors and/or particular players in the territory, a tendential standardization reducing any cultural and/or environmental specificity, and the definition of exogenous development trajectories scarcely consistent with local "territorial desires" [34] or expectations.

Exogenous capital interventions have been relevant, of course, in the period of the Italian economic boom, ensuring the survival of agriculture in the most peripheral and depopulated rural areas of the country; however, the evident top-down approach consists, first of all, in the dispersion of the locally-based added value of traditional rural communities, impoverishing and dangerously reducing their self-confidence and capacity-building abilities.

An endogenous approach, on the contrary, frames rural development processes in a systemic form and with a territorial perspective at a regional or subregional scale. By doing this, policy actions orient communities to participate in the conservation and vakorisation of their "territorial capital" [35], carefully modulating interventions to the needs and perspectives of local populations [36].

In the processes of endogenous development, the role of agriculture still remains pre-eminent [37], although in decline, due to (or by virtue of) the widespread outsourcing of local employment, by making the economies of smaller rural contexts increasingy similar to urban ones, somehow even loosing their peculiarities, such as for example the practices and habits linked to the traditional production and consumption of food.

The contrast between endogenous and exogenous development has, however, been heavily criticized, both because it is dichotomous and above all because it tends to ignore, underestimate, or remove the conflicts connected to endogenous development [38], which reduces its economic and social potential.

Even the most widespread and celebrated form of endogenous rural development successfully tested in the last thirty years in Europe, namely, the LEADER program, has been demonstrated by various scientific studies to have limits in terms of efficiency resulting from participation deficits or phenomena of elitism [39–41], which in some cases lead to the failure of the networks underlying the LAGs, and therefore of the same actors delegated to implement these approaches [42–44].

It should be noted that the conservation of the explicitly exogenous nature of the other policies complementary to those of rural development, such as the first pillar of the CAP or the pseudo-endogenous action of regional development (POR), further limits the effectiveness of the endogenous approach, proposing answers apparently consistent with the needs of the local population through the selective involvement of local actors in order to privilege (or protect) the interests of the most powerful social groups [45].

It therefore follows, as Brunori and Rossi [46] underline, that rural contexts tend to assume configurations influenced by external pressures even if based on local resources guided by endogenous actors, which, however, tend to reflect the logics and dynamics typical of economic globalization. Therefore, as Ward et al. [47] note, the endogenous model is not very realistic and hardly practicable except through a hybrid approach, defined as "neo-endogenous" (cite the most important works), which considers the local contexts as more extended towards the outside through the activation of dynamic interactions with the wider economic, political, and institutional environment in which they operate. As Ray [48] observes, in fact, development based on endogenous resources is not always attributable to the exclusive action of local actors, and is very often activated and/or corroborated by exogenous interventions from above, through policies aimed at activation of participatory processes by intermediate actors such as NGOs, which consider the activation of territorial development as a tool to achieve their own institutional objectives. Neo-endogenous development is therefore the result of the combination of sources and forces of different origin (intermediate and/or higher) concentrated on the activation and enhancement of the endogenous resources of the territories for the purposes of their economic and social regeneration.

The success of these processes therefore depends on the resources and cultural heritage of the communities as well as on the competence and awareness of political actors to recognize and make use of certain endogenous development practices ("acting from above to facilitate action from below") whose initiative could trigger virtuous processes of normative isomorphism [49]. This is what Bock [50] calls the nexogenous approach, emphasizing the reunification of the various forces acting in the same rural space in order to evoke the need for the (re)construction of solid links between the political and social components of each community.

The center of our work, therefore, is to reconstruct the genesis and dynamics of the regenerative development processes experienced by three small Italian rural communities (of Northern, Central, Southern Italy) which we could define as local economies in

transition, focusing on the possibility of these being configured as different expressions of the neo-endogenous approach. The recent experimentation with alternative agricultural and rural models has in these cases activated, in a more or less conscious way, new forms of safeguarding and enhancing bio-cultural heritage such as ecomuseums, responsible purchasing groups, cultural animation associations, and community cooperatives, to name a few.

The added value of an ethnography based on Italian cases is due to: (a) the substantial homogeneity of national/institutional policies with respect to local development; (b) the possibility for an insightful comparison among different responses and community-involvement levels in different macro-areas of the same country; (c) a particularly challenging present phase where several programs and vigorous actions of local/rural development have been launched and implemented to cope with post-COVID National Plans of Recovery and Resilience (PNRR), as well as the restarting of other impactful development programs.

Key topics for our interpretation of the three fieldwork areas include roots, rhizomes, networks, and meshworks. These are all concepts referring to an alternative imaginary network of local territories, enabling changes and adaptations to present conditions and addressing critical issues on climate change impacts on agriculture or the increasingly extractive market and predatory distribution of agri-food products. Through these pivotal interpretative keys, we try to observe how local populations manage to live and remain in territories hitherto condemned to obsolescence, starting from purely mercantile logic and today rhetorically recovered as new places for urban refreshment and escape from the rhythms and pressure of crowded urban spaces.

## 2. Materials and Methods

### 2.1. Materials

The present research followed and especially focused on specific data, records and materials largely recollected and sedimented in the framework of the first Online International Course of the project EARTH [51], including:

1.  In-depth interviews were conducted with privileged witnesses in each of the three elicited cases, namely, Stefano Martini, President of the Ecomuseum of Pontebernardo; Lino Gentile, Mayor of Castel del Giudice; and Chiara Vacirca, Member of the Collective group coordinating the Association "Casa delle Agri-Culture" of Castiglione d'Otranto.
2.  A medium-long period of ethnographic/grounded observation (random interviews, informal speeches, media/new media narratives, video documentation, official documents, critical analysis, and more); in Castel del Giudice, research has ongoing for fifteen years, allowing long-term observation of the processes implemented. In Pontebernardo, as in Castiglione d'Otranto, a less intensive, though almost daily, monitoring and research was begun in 2018.
3.  Several refining interviews allowed us to deepen our critical reflections on the cases.

In Pontebernardo, the interviewees were (a) the President of the local LAG; (b) the responsible member of the Mountain Union; (c) the referee of the cooperative that manages the Ecomuseum's services; (d) two members of local associations and enterprises; (e) the Mayor of Pontebernardo, who is additionally engaged in interesting entrepreneurial activities in the local territory.

In Castel del Giudice, interviewees were (a) the responsible member of the agricultural company 'Melise'; (b) the referee of the new brewery 'Maltolento'; (c) the responsible member of the ex-SPRAR project; (d) a municipal functionary engaged in several actions of programming and fundraising for the municipality.

In Castiglione d'Otranto, interviewees were (a) the referees of the local community mill; (b) the coordinator of the activities and programming of the Green Night Festival; (c) the referees of community vegetables and fruit production; (d) a representative of a cultural association connected to the Green Night Festival; (e) several local citizens and stakeholders participating and interested by the specific actions developed in the three

addressed contexts, especially the "rural open parliaments" organized on the occasion of the festival itself.

All interviews were based on a semi-structured model, with several basic starting questions followed by more defined questions adapted to the development of the first part of the encounter. Several of interviews were conducted in person; only few had to be repeated or continued online. Others had to be conducted entirely online due to the social distancing imposed by the COVID-19 pandemic. However, the use of different survey criteria imposed by the particular pandemic conditions did not imply significant problems of involvement and/or participation of the actors nor of detection and interpretation of the data, thanks to the author's long-term and deep awareness of local contexts [30,31,51].

All of the interviews conducted online involved local stakeholders who the authors had already met and who were well-known from previous observations and field visits.

The choice of these three cases was connected to three specific and consistent reasons:

1. All three cases were elicited and presented/discussed during the OIC 1 (First Online International Course of the project EARTH), ensuring wider discussion;
2. In all cases, there was a connection with the Centre of Research 'BIOCULT' for 'biocultural resources and local development', a centre in which two of the three authors participate;
3. The representative role each of the cases has in the three main macro-areas of Italy;
4. The presence, with different shades and postures, of similar socio-cultural and economic-political actors (i.e., the specific and particular protagonism of certain individual local actors such as the mayor, principal organizers, experts, etc.), which enabled close multi-actor observation for a more aware comparative approach.

*2.2. Methodology*

The methodology adopted was based on qualitative research, essentially inspired by ethnographic tools, attitudes, and timing, as well as on the comparative evaluation of case studies concerning three experiences of territorial regeneration in three different Italian regions located, respectively, in the north, center, and south of Italy (Figure 1):

- The Eco-museum of Pastoralism in Pontebernardo (Province of Cuneo, Northern Region of Piedmont)
- The Municipality, local agro-ecological industries, and cooperatives of Castel del Giudice (Province of Isernia, Central-Southern Region of Molise)
- The local community and association "House of Agri-Culture" in Castiglione d'Otranto (Province of Otranto, Southern Region of Puglia).

These cases, although referable to three different geographical, economic, and social contexts, are animated by the same motivations and similar results while occupying different positions within the gradient defined by their differences in terms of endogenous and neo-endogenous approaches to rural development.

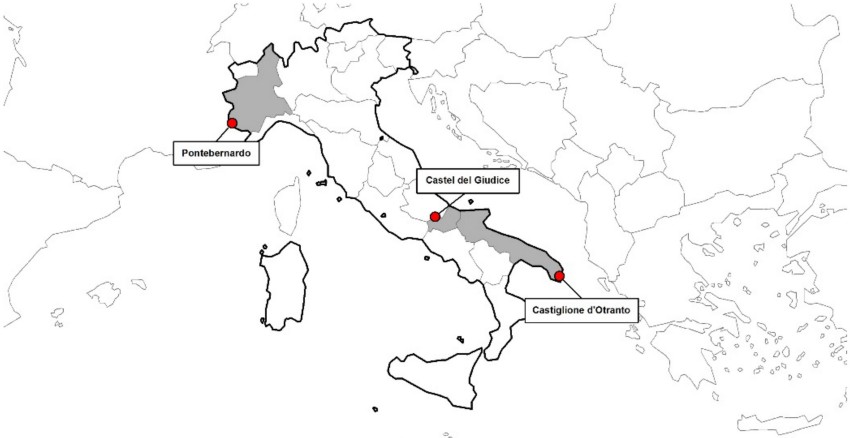

**Figure 1.** Geographical location of case studies (Map by Giuseppe Pistacchio).

The present case study-based approach ensured the maintenance of a systemic perspective in each regional context and allowed for in-depth detection of problems associated with the productive and/or organizational decisions of each, at the same time providing greater effectiveness with respect to the quantitative approaches.

Qualitative methods, in particular the set of ethnographic investigation and observation techniques, contribute in a particularly relevant way to investigations into social entrepreneurship [52,53], benefiting from the direct presence of the researchers in the field that allows exploration of the specific social dynamics underlying the heterodox and successful initiatives that the study intends to analyze.

Following the scheme of, among others, Chatzichristos et al. [54], the present work is inspired by constructivist Grounded Theory, which is usually based on the direct participation of the researchers in the construction of the data, starting from the experience of the main actors of the investigated process by means of the observation of their actions and the systematization of their experiences. The construction of the data used in this work therefore stems from the patient work of observation and interaction between the authors and the research material provided by the three territories that are the subject of this analysis. Alongside the grounded theory approach, other methods of participant observation, namely, multi-situated deep mapping and multivocal critical ethnography have been included; these are part of an overall ethnographic and deconstructive approach.

A set of innovative methodologies was added during the research and project development:

- A case study-based analysis developed, refined, and tested during the Online International Course of the project EARTH as a research tool as well as a powerful learning strategy;
- Quantitative insights as a clarifying integration and schematization of such a rich grounded analysis;
- Focus groups, as a way to develop and implement community participation and a multi-actor research perspective;
- Interactive meetings, experimented with during the first Online International Course of the Project EARTH as a challenge to the present impossibility to be concretely on the fieldwork;
- Affective and network-building hotspots as a powerful tool of community mapping for defining and deepening the sense of place, awareness and definition of belongingness, and consequent involvement in local development processes;
- Analysis of multi-scale interactions in the rural contexts where sociability and cooperative networks among local stakeholders are developing (i.e., the Castiglione d'Otranto case, in particular, observed especially in the last three years with particular reference to the 2021 Summer edition);
- Senso-biographic walks (particularly one with the Mayor of Castel del Giudice during the EARTH Project Online International Course of November 2020, which helped to develop a form of embedded narrative about the spaces and a reformulated concept of neo-endogenous opportunities for rural development);
- Micro-phenomenological analysis and local mapping, as well as the observation of innovation hubs (meetings, social encounters, fairs, etc.), especially observed in the context of an eco-museum proposal at the Pontebernardo in the Piedmont Region, where a quite successful eco-museum of Pastoralism has been set and developed during the last two decades and helped to consolidate local/regional/national and international networks.

The selected cases can be traced back to three different processes and forms of 'heritagization' of community environmental, cultural and intangible resources based, respectively, on the reinterpretation of pastoral practices and transhumance, on the diversification of agriculture combined with social innovation, and on the community recovery of rural structures and local peasant culture.

1.　A comparative effort was set up among the three cases, based on five main items/notions defined and discussed among the authors:

　　a.　Biocultural heritage impact and concerns;
　　b.　Old and new rurality re-definitions throughout the cases;
　　c.　Return and repopulation processes to inner/fragile/mountainous/rural regions;
　　d.　Social Innovation in agriculture/smart agriculture and peasant social inclusion;
　　e.　Participation and participatory inventories, public engagement in the rural public space, and multi-actor involvement (Table 1).

**Table 1.** Thematization of empirical analysis.

| Theoretical Factors | Thematization of Empirical Analysis |
| --- | --- |
| Biocultural heritage, embeddedness, grounded theory | Impact of knowledge–practice systems, sense of place in the definition of rural territorial capital |
| Rural development models | Actors, resources, and shared values, goals and development trajectories |
| Resilience, social innovation, inclusiveness | Reduction of depopulation through biocultural heritage valorization and innovative and inclusive rural practices |
| Regeneration | Return and repopulation processes through diversified agricultural activities |
| Participation | Multi-actor involvement, participatory inventories, public engagement, and social innovation practices |

## 3. Results

### 3.1. Castel del Giudice

Castel del Giudice (CdG) is an Apennine town in central-southern Italy. With just over 300 inhabitants, it is among the smallest communities in the Upper and Middle Sannio, one of the four inland areas of the Molise region. CdG was affected by significant demographic decline (−61% between 1961 and 2021) as the local population aged, although the economic and social innovation initiatives illustrated below have managed to effectively reduce this phenomenon (−13% between 2011 and 2021).

Agricultural activity, despite having registered a profound reduction in both farms and UAA, respectively −92% and −70% in the inter-census interval 1982–2010, continues to be fundamental for the local economy and the fulcrum of the regeneration process thanks to the recognition of its ecological and social potential for release through a radical innovation process (i.e., conversion to organic farming) and managerial reorganization of the same.

However, among the administrators and residents there are growth expectations aimed above all at increased food and wine and experiential tourism, counting on the externalities connected to some considerable attractions in the neighboring municipalities (Agnone and Pietrabbondante) as well as on the proximity to skiing areas.

Empirical research has made it possible to characterize the development model adopted by CdG in the last twenty years as consisting of an interesting process of social innovation connected to the development of economic initiatives the overall evaluation of which offers the opportunity to position this case on the gradient defined by the deviation of the neo-endogenous approach to rural development from the truly endogenous.

The awareness of the serious problem of displacement and the geographical and political periphery of the municipality with respect to the center of regional political action is firmly linked to welfare development practices. Such vigorous development action is connected to exogenous development coupled with a young municipal administration, established in the 1999 and led by a mayor (who is still in office) determined to activate and identify the resources to start an urgent project of structural, economic, and social regeneration to ensure the survival of the municipality.

With the demographic decline overt, most residents were elders and job opportunities very scarce. The local administration decided (to what degree this was a conscious choice is not clear) to intervene with respect to one of the symbols of the small community, the

school building, desolate and in disuse at that time and which represented only a cost for the nearly-empty municipal coffers. At that time, an urban planning solution was sought to ensure and accelerate the construction of one of the health centers planned in a previous project of the regional government. This was a social welfare residence (RSA) which, although consistent with the needs of an increasingly elderly population, seemed destined to undergo insurmountable bureaucratic delays and difficulties. Failure to build the residence, which in the meantime had been sited in the old school building, would have prevented the recovery of a property that once represented the vitality of that community, and would have denied the residents a service of indisputable usefulness and a certain centrality with respect to neighboring municipalities.

The problem was solved when an entrepreneur extraneous to the community (originally from the neighboring municipality of Capracotta), who in those years had set up a new factory of his company, a leader in precision mechanics located in Lombardy, expressed to the municipal administration his willingness to participate in any business initiatives aimed at social objectives. A new mixed public–private company was therefore set up which took on the huge investment necessary for the realization of the RSA health residence. The decision was made following participatory meetings between the administration and the residents, several of whom decided to assist by financing the project directly.

The fruitful interaction between the municipal administration and the owner of the mechanical company for the construction of the RSA offered the CdG community an important opportunity to reflect on the availability and forms of reuse of their common resources in the context of recovering the broken relationship of the village with the school, perhaps the most symbolic place of memory in the small community. This occasion offered the opportunity to reactivate social relations, substantiated in manifestations of consent/dissent and/or mutual consultations for the assessment of the risks and opportunities of a possible direct financial involvement of each member of the community in the project admitted by the scheme adopted by the municipal administration.

The collaborative and participatory attitude experienced in this circumstance by the majority of the residents induced reflective attitudes that made it possible to recover local secular traditions of solidarity and collaboration which were definitively shared by the local population.

These participatory practices were therefore replicated in the establishment of an agricultural company, the "Melise", specialized in the production of apples and organic preserves (apple in Italian is "mela"), a name evoking a link with the territory ("Molise/Melise") and underlining the commitment to economic development and social regeneration of the CdG community. In this case, many residents participated in the company's capital and others became employees. The fruitful interaction with various external actors made it possible to develop new professional skills over time both in the field of agronomic practices and in the managerial and commercial areas, experimenting with increasingly integrated forms of cooperation for the enhancement of products through the supply of solidarity buying groups such as GAS (Gruppi di Acquisto Solidale) in the nearby urban contexts of Naples and Rome.

Thus, agriculture returned to the center of CdG's development project. As demonstrated by the recent further development of the company engaged in a process of productive diversification in order to make the best use of the human resources available, considering that in the cultivation of apple trees there are dead periods, Melise has tried spelt and organic barley, functional to the strengthening of the agricultural brewery for the production of beer with a local brand "Malto Lento". In this sense, Melise is trying to enhance the cultivation of hops, which spontaneously grow in the area, and to support community cooperatives such as a community apiary, providing a pollination service.

The same methods, by now consolidated, accompanied the recovery and redevelopment project of "Borgo Tufi", a semi-abandoned area transformed into a comfortable widespread hotel. In this case, however, substantial differences were found in the degree of sharing and appreciation of the result by the community. The construction of the hotel

has used and enhanced a space that once housed the town's stables, its productive heart and anciently considered by their small owners as a symbolic center of the community that survived abandonment and ensuing decay.

In certain cases, in the course of our field activities a nostalgic regret for the urban transformation that had been approved by the community itself emerged alongside appreciation for the transformation of the village into a widespread hotel. Moreover, the size and methods of the investment determined that only two private entities were partners with the Municipality of CdG, thus reducing the participatory impact, which in the two previous projects had been larger and more effective.

The common theme to these three interesting examples of the social innovation in Castel del Giudice is that a participatory approach was fundamental to creating the respective public–private social enterprises. Each was set up in partnership with the municipality and the mayor, the local community, and other key sectoral stakeholders in the agricultural, social care, and tourism fields. The Castel del Giudice case, therefore, is significant in terms of social innovation because it illustrates how a tiny and remote rural municipality facing significant demographic challenges, with an ageing population and limited development opportunities, can develop a proactive local development strategy. Many nearby administrations are trying to follow the example of Castel del Giudice without achieving the same outcomes. In any case, this project of extending the practice of Castel del Giudice to the nearby villages has found institutionalization in the LAG "Alto Molise".

However, several interviews raised issues, a source of minor dissent in the community, linked to the speed at which the development project moved forward and to the impossibility of fully understanding what was happening in the community or the beneficial aspects for the village. The perception was rather of being threatened by a process of expropriation due to inability to participate in a process that effectively involved the entire community. For a period, it was possible to individuate positions that somehow were in open disagreement with the mayor and the municipal administration, with a part of the community not actively participating in the celebration of the feast of apples. This dissent seems to be recomposed during more recent years with the acceleration of development projects that is attracting young people from the nearby villages through the creation of job opportunities based on a coherent set of projects based on enhancement of local resources.

### 3.2. Castiglione d'Otranto

The Municipality of Andrano is located within the area of the Costa Otranto-Santa Maria di Leuca and Bosco Tricase Regional Natural Park.

The population in 2019 was 4.666 inhabitants; the resident population over 75 years old was over 12.5%, of the total while that under four years old was 3%. In Castiglione d'Otranto, the population was estimated at less than 1000 inhabitants.

The Casa of Agri-Cultures 'Tullia e Gino' association was founded by a group of villagers who migrated outside of the Apulia Region for school or work and were determined to build the social, cultural, and economic conditions to return to living in their home village. In the public discussions around the Green Night, as well as in personal interviews, there was a recurrent reference to the concept of "restanza" (remaining), coined by the anthropologist Vito Teti [55,56]. In this vision, remaining or returning to stay is not a matter of weakness; rather it is one of courage. This approach is a novelty, inverting the situation of grandfathers; young people feel that there are new opportunities based on new lifestyles, a different, sustainable, and profound way of experiencing rural life and enhancing the area without commercializing it. There is the possibility of remaining in the villages without being forced to emigrate elsewhere and without having to undergo processes of new colonization by the large economic powers.

In order to disseminate this new vision in the community, the association created an agriculture cooperative, a community mill, the annual event called the Green Night, and other concrete actions realizing their vision, including a special focus on under-35 villagers.

The traditional concept of rurality was rooted in subsistence agriculture based on drudgery and self-exploitation of peasants. Thus, agriculture was associated with back-breaking work and exploitation for thousands of Salento farmers, while rural life was consecrated to a poor cultural life without access to basic education and services.

In addition to extensive olive culture, up to the 1990s agriculture in Salento consisted of growing tobacco through cooperatives with strong subsidies from the state and high levels of exploitation of land workers, particularly women.

After 2011, the association's action and the organization of the Green Night have played a central role in the reconfiguration of the imagery of agricultural practice based on agroecology, multifunctionality, bio-cultural heritage, and environmental sustainability. Beginning in 2010 the bacteria *Xylella fastidiosa* has infected most of the olive trees in Salento, creating a spectral landscape, as the infected olive trees have to be uprooted, destroyed, and practically erased from the Apulian cultural scene. This situation has led to a further abandonment of land and agriculture, with many farms being forced to close.

The association approached the discussion acknowledging the limits of monoculture farming based on large amounts of pesticides and herbicides and providing an alternative to planting a new variety of olive tree that is resistant to *xylella*: the diversification of the cultivation of traditional varieties of cereals and legumes using organic methods. This approach constitutes a strong innovation in a context that has been characterized by the intensive use of chemicals in agriculture.

The association formed a cooperative to manage farming activities covering about twenty hectares of cereals, tomatoes, and vegetables, which are partially processed locally and mainly sold at the local level through the People's Purchasing Group (GAP), the shop of the Community Mill shops and local restaurants, as well as in the rest of Italy through a network of small grocery stores.

In this framework, the Community Mill serves as an infrastructure supporting wheat production and processing while at the same fostering an innovation in social dynamics among farmers by providing an important meeting place to exchange knowledge and share problems with their agricultural activities.

Moreover, external farmers have been involved in activities as the collective purchase of seeds, cultivation using natural (organic) methods, land rotation, the introduction of pollinator plants to stimulate the return of bees to the land, and the purchase by the cooperative of external production at a fair price.

The Salento People's Purchasing Group was born out of the need to bring farmers and citizens into closer dialogue. Formally, it is a group of consumers who decide to "shop" together by going directly to the producers, breaking the chain typical of large-scale organized distribution, i.e., the one that passes through wholesalers and shops, pushing up final prices. The GAP has the advantage of creating a double network: the first is that of citizens who decide to opt for collective purchases, ensuring lower costs and genuine food; the second is that of selected Salento micro-producers who comply with a precise ethical protocol, guaranteeing a supply chain that is both short and natural. These principles are monitored by the Casa of Agri-Culture's 'Tullia e Gino' association. The Popular Purchasing Group serves to facilitate above all the purchase of basic necessities, all of which come from natural agriculture and farming. The products in the 'basket' include fruit and vegetables produced without the use of chemicals by local farmers, soft and hard cheeses made by master cheesemakers who use milk from animals reared in a non-intensive manner, flours, cereals and baked goods made from spelt and ancient grains grown locally without pesticides, certified hemp products, typical pulses from the community cooperative in Zollino, wine from untreated vineyards tended by Salento winegrowers, and food products made from natural products.

The Casa of Agri-Culture's 'Tullia e Gino' association promotes social change through socially engaged art and culture.

The Green Night is a major Apulian event dedicated to building another idea of territory, far from the exploitation of natural resources and consumerism and closer to reflection on how to enhance common goods, small economies, sustainability, and minor communities. The Green Night takes places at the end of August at the end of the tourist season in order to target the participation of the local community and local authorities. The five-day program includes concerts, a market of local products, rural parliaments, and public debates with participants from all over Italy. It is a very successful event and participation has grown over the years to reach a peak of 30,000 visitors.

The "Luigi Russo" Inclusion Nursery functions as a community social laboratory for the production, selection and conservation of seeds and mother plants of local cultivars of vegetables (cabbage, fennel, tomatoes, courgettes, aubergines, etc.) and minor fruits. The nursery aims to combat involuntary loneliness, especially in the elderly population, migrants, and people with disabilities through initiatives and paths of participatory involvement such as support for activities accompanying the weaker sections of the population to work and activities to promote and safeguard environmental areas. The nursery serves to maintain the usefulness of genetic resources and to protect the rights of farmers; plants at risk of extinction are catalogued and reproduced in order to increase agricultural biodiversity with collective sowing involving the community, a library of saved seeds has been set up, a small greenhouse for the care of horticultural plants has been be built in bio-building, and ethnobotanical cooking and bread-making workshops are active.

In addition to the recovery of bio-cultural heritage through community involvement, the Casa of Agri-Culture's 'Tullia e Gino' association has launched a community Agri-ludoteque project, a playful and inclusive space dedicated to children and focused on safeguarding the knowledge, practices and values of the rural world lived in a new way. The project seeks to respond to the lack of a primary and secondary school; the school has been closed for years due to low birth rates, and this is becoming a disruptive factor for families, who must enroll their children in other municipalities (Montesano, Andrano, Tricase), weakening community ties in a town that is depopulating. This is a difficult hurdle to overcome in the inclusion of children who have other abilities or are at social risk (i.e., adolescent and poorly educated parents, unemployed fathers, children of migrants, school evasion, etc.), all situations present in the area.

The Agri-Ludoteque wants to ensure stable pedagogical and socialization activities for children, with the primary aim of including children with autism spectrum disorders, as well as to rebuild a sort of "school group" among families and to shore up education on issues of environmental sustainability and respect for others.

Moreover, the Casa of Agri-Culture's 'Tullia e Gino' association hosts artists from all over the world with such socially engaged actions as planting trees, building dry stone walls, creating new signed routes in the countryside, and creating murals in the city park.

*3.3. Pontebernardo*

The Ecomuseum of Pastoralism is located in the small village of Pontebernardo (80 inhabitants), part of the Municipality of Pietraporzio in the Province of Cuneo, Italian Region of Piemonte. It is a mountain village (1246 m above sea level) located in the Stura Valley, in an area historically characterized by the presence of transhumance routes between the Italian mountains and the French lowlands, an area in which the herds remained grazing during the rigorous valley winters. The headquarters of the Ecomuseum occupies a building in the center of the village that was acquired by the Mountain Government Authority and has been renovated to make it available for community activities. The eco-museum center of interpretation hosts a small dairy managed by the shepherd families who live in the area to prepare an excellent cheese, the "Toumo of the Ecomuseum", and a plant for the processing of Sambucana sheep meat where excellent sausages are produced.

In the same building there is a tasting point inaugurated in 2008. The Center is equipped as a station for breeding rams, managed by "L'Escaroun", the breeders' consortium. A center called "Na Draio per Vioure" completes the knowledge–practice systematic restitution of the

ecomuseum, offering to visitors a broad overview of sheep farming with a substantial portion related to the recreation of local reality in its most direct reference to products and flavors. This space houses a newly-installed modern shop for the purchase of precious Sambucan sheep wool products and a multifunctional room for Ecomuseum activities.

At the end of the last century, the Cuneo valley area was one of the most affected by depopulation and the rapid deterioration of most of its traditional production activities, mainly sheep farming and related production such as cheese- and sausage-making and handicrafts. The incidence of tourism was largely limited compared to the urban centers of the region, Turin in particular, or other areas related to ethno-tourism, for example, the Barolo and Langhe areas. The creation of the ecomuseum of pastoralism emerged in the 1980s mobilized by the mountain community, which managed to recover traditional grazing activities and added value to the pastures historically present in the area, as well as native sheep breeds particularly adapted to the territorial conditions where these communities are settled.

Piemonte Region presented the first regional law on ecomuseums in 1995; in this area of the province of Cuneo, however, various experiences arose much earlier to safeguard and improve the pastoral practice deeply rooted in this territory. The Ecomuseum of Pastoralism particularly aims to raise awareness of the importance of pastoral culture in the area, generating actions to safeguard and improve the structures, landscape, and set of practices related to transhumant grazing and promoting the revitalization of the production of purebred "sambucana" sheep. The Sambucana sheep has been raised from ancient times in the Valle Stura di Demonte, where this practice has always been the most important activity that has allowed the extensive pastures of the highlands to be exploited. It is a particularly rustic animal and suitable for the environment in which it lives; in fact, it spends the summer grazing period in the rocky and steep mountains, with very low temperatures in the spring and autumn months and at night. In the winter it remains in the stables, feeding only on hay produced on site. The Sambucana sheep is an excellent producer of meat, milk, and wool.

Within the framework of the recovery process of pastures and pastoral activity, the Ecomuseum constitutes an initiative strongly supported by the innovative and visionary Law of Piemonte. This regulation has inspired and mobilized various activities, among which the following stand out:

— The recovery of the Sambucana sheep;
— The revitalization of grasslands along the Routo, the old road leading from the valley to the plains of the French Crau;
— The production of Sambucano cheeses, lambs, and their derivatives;
— The use of locally produced wool for the manufacture of clothing for trekking;
— Research on genealogical lines, productive activities and the development of the inhabited areas of the Valley, which influenced the historical recovery of experiences and memories as well as the preparation of reports and the proposal of local initiatives to promote the area;
— The recovery and enhancement of the typical landscape of the Valley associated with the elements of pastoral activity (houses, cabins, corrals, stables, grazing areas, residential areas, dairies).

Our visit (accompanied by Stefano Martini, the referent and president of the ecomuseum of pastoralism) to the ecomuseum hotspot and the territory in which the ecomuseum's activities are interested highlighted the influence that this experienced group of local actors had in soliciting the active participation of the local population and in enhancing territorial revitalization. The same view is clearly set and disseminated throughout media sheets and social network advertisement about the Stura Valley tourist attractions.

The Ecomuseum of Pastoralism of Pontebernardo had a pivotal role in leading the process of local regeneration, starting from a powerful and efficient public/private cooperation, institutional and community shared actions, and the support and care of associative and informal actors. Such circular and proactive cooperation of all the local stakeholders in

the territory has ensured the continuity of the project despite the reduction, to an extent, of the economic support for the ecomuseum and several difficulties linked to changes in territorial policies, LAG strategies, etc.

The Municipality of Pietraporzio continued, as an institution of territorial proximity, to work on actions oriented at recovery processes and revitalizing special local productions. The "Escaroun" Consortium contributed through the enhancement of native livestock sheep breeding and with efficient branding of local production, especially focusing on the cultural heritage connected to the conservation and valorization of the special elements of local biodiversity. "Escaroun" is a term in the local dialect which refers to a sheep that separates from the herd to search for the best pastures; locally, such sheep are considered the best thanks to their ability to identify the pastures with the highest quality.

The Mountain Community of the Stura Valley (CN), which currently forms the Union of Mountain Municipalities, has led several projects to promote and reactivate productive activities and lines of European financing, including the recovery of production of the Sambucan sheep and its support as protected livestock within the European/National framework.

The Piemonte Region, in particular the Regional Ecomuseum's Office for Culture, has recently re-empowered the action of ecomuseums through the elaboration of a new version of the Regional Law and by supporting, along with other regions, the opportunity to elaborate a unique National Law on Ecomuseums.

The University of Turin, Department of Agrarian Studies, Masters in Alpine Cultures has for many years provided knowledge and policy proposals for the recovery and valorization of mountain activities in the area, building relations with other institutions and promoters such as the Maison de la transhumance en la Crau in France, Rete APPIA for pastoralism, and other Universities in Italy to revitalize traditional and transhumant grazing.

These Agricultural Technical Schools (through educational laboratories, field visits, etc.), along with cultural associations, have collaborated on promotion of the territory (through musical groups, local, walkers, artists, etc.) for tourism.

This cluster of several different local actors has supported the process of development and consolidation of the Ecomuseum of Pastoralism and its present major outputs. Interviewed during the first Online International Course of the project EARTH organized by the University of Molise, the President of the ecomuseum, first of all, pointed out the resumption of the breeding of the Sambucana sheep and the elaboration of derived products with traditional and sustainable production methods as the pivotal action realized by the ecomuseum.

At the same time, Miriam Rubeis, adjunct researcher and officer of the ecomuseum, affirms that its activities are aimed at raising awareness, knowledge, socialization, and dissemination of local traditional history and culture through the elaboration of tourist routes, educational workshops/didactic farm proposals, animation activities for the local population, publication of a scientific magazines, and the creation of brochures and illustrative guides.

An important element of the local regeneration process is connected to specific craft activities linked to the territory supported by embedded sustainable development programs. In this framework, characteristic local enterprises such as a small company dedicated to the cultivation of aromatic and medicinal herbs, a soap factory, a dairy production plant, an establishment where sheep meat is processed, and a hotel, a tourist proposal for trails and visits to characteristic places in the area have all emerged, based on knowledge of the territory. In these regenerating actions, the role of the ecomuseum has been coupled with the Mountain Union as well as with the LAG's actions, forming a strong nexus between local cultural heritage and social innovation processes in terms of the creation of small cooperatives and consortia.

## 4. Discussion

In the course of several interviews, the ethnographic observations, debates, and focus groups developed before, during, and after the EARTH Project Online International Course as well as the impact of biocultural heritage in these different processes of local regeneration and social innovation were tested and discussed based on at least three key concepts:

(1) rural change and its implications in terms of maximizing opportunities for inhabitants while managing risks;
(2) the shared management of local heritage and resources implying different degrees of collective participation in the regeneration processes;
(3) a more integrated approach to rural development, with an innovative relationship to public administration creating new forms of public–private partnerships.

This new scenario implies a reflection on common goods developed beginning at the start of the new millennium [57–60]: a reflection based on concepts such as reciprocity, gifting, degrowth, and alternative and fundamental economies. The observed community practices show a unique intent of sustainable rural development and a new moral economy centered on circularity, exchange, and cooperation both as apparently anti-economic categories and as effectively anti-commodifying and anti-reifying social "gift" practices [61]. Such reciprocity and circularity is oriented towards making spaces inhabitable that would be considered uninhabitable according to urban-centric late modernist logic.

In the specific case of Italy, anthropologists and social scientists have more recently discussed this process reflecting on the concept of *restanza* [55], a concept that was consciously mentioned by several of the interviewed witnesses as part of their inspiring notions. The concept of *restanza* defines the position of those who decide to stay, renouncing the severance of their link with their land and community of origin and replacing resignation with a proactive attitude. The choice to come back or to stay is a drive tending to the construction of a new polis, a new way of living and organising spaces, economies, relationships, something approximating to the birth of a new community. The word *restanza* (similar though distinct from 'resilience') denotes a creative and dynamic act, a link to the idea of moving even while remaining in the same place. To stay, in fact, requires the ability to relate past and present, to re-actualise lost and viable routes.

In this way, communities show ambivalent and critical movements in their development strategies that deserve specific final considerations.

In Castel del Giudice, for example, concerning the fields of agriculture, tourism and health assistance there is not always an exclusively common vision; rather, the result of different opportunities, of which the agricultural axis associated with Melise was not necessarily leading the others. Twenty years of practice led to the building of a coherent vision and narrative mainly driven by the practice of the organic agriculture of Melise, which provided a particular meaning of sustainability to the opportunity to enhance local resources embedded in the development project of Castel del Giudice.

For instance, this approach was extended to Borgo Tufi as the showcase of the development project of Castel del Giudice. The renovation of the village and the colour plan in order to create continuity with Borgo Tufi from the aesthetic point of view, has to do with the fact that Borgo Tufi and Castel del Giudice seem to be two contrasting realities. It can been seen that on the one hand, paradoxically, in the older case it has been possible to make an important renovation, and it is just as important to do this in the part where the citizens still live in order to provide a sense of continuity between the two realities. At the moment, in fact, tourists from Borgo Tufi rarely seem to go to the modern nucleus of Castel del Giudice, and the Castellans never seem go down to Borgo Tufi. Moreover, Borgo Tufi, rather than a luxury resort oriented toward the nearby ski slopes, is reorienting towards an experiential tourism that has a great deal to do with a relationship with the community that provides services (e.g., traditional breakfast prepared by local ladies, attending evening meetings at the bar, etc.).

This new approach is embedded in the new Food Local Plan, which places food system as vital and central to the production system everywhere. Food is therefore considered having a fundamental role in the quality of life of the population and is increasingly recognised as a key issue in socioeconomic and environmental policies to reinterpret and reconstruct local production and consumption processes, thus contributing to the proper management of natural and social capital. In fact, food has close interconnections with urban public policies and is a cross-cutting issue that affects and influences the local economy, public health, and the quality of urban, peri-urban, and rural areas. An increasing number of cities are beginning to consider food as a key element around which to plan integrated and sustainable urban development. The actions implemented in this way enable the local food system to be used for the management of economic, environmental and social priorities, guaranteeing a coordinating function with respect to policies and projects in different thematic areas that can be designed and implemented independently of each other.

Therefore, food represents a strategic lever for achieving sustainability across the board by putting it into circulation in the form of matter and energy and disseminating its value in terms of environmental, social, and economic impact.

The Food Local Plan was presented and approved by the Municipality in 2019, defining food as a strategic lever and designing a coherent policy framework building on the experience of Melise. In doing this, organic agriculture and local food production are driving other practices such as the RSA, SPRAR/SAI, and Borgo Tufi towards a definition of a Green Community which promotes healthy and sustainable lifestyles, including plastic-free, carbon-free, and pesticide-free territory and experiential and rural tourism activities aimed at diversifying and expanding the territorial tourist offerings, which are otherwise subject to seasonality.

These convergences of the different practices in the experience of Castel del Giudice under a common narrative and strategy of territorial marketing starting from food production and consumption provide a clear perspective to the development trajectory of Castel del Giudice for the next decade. The challenge, however, is that planning sustainable polices in rural area is not easy. The high level of dependency on surrounding urban centres for jobs and services needs to be recognised. At the same time, the need to find new job opportunities is central as well as the role of the third sector in addressing social exclusion for the elderly and providing access to essential services. Ultimately, as several interviewees argued, social innovation needs to involve and gather the views of local communities, businesses, and landowners regarding the key issues for local development and to agree on what 'reasonable' levels of access and basic 'local' services should be provided or aimed for.

In summary, Castel del Giudice's social innovation illustrates a new model of how to manage natural and rural resources as well as environmental heritage through a strong innovative approach in public administration, creating new opportunities based on the intersection of different tools. This is noteworthy because hitherto in Italy and the Molise region the development of public–private partnerships has been relatively limited and there are limited good practice examples to learn from. The Castel del Giudice case, however, represents a good illustration for other small and remote rural locales to learn from. In fact, there has been a great deal of national interest within Italy about the Castel del Giudice story, and even internationally. This has significantly raised the profile of the municipality, which now regularly hosts practitioners from other places in Italy and abroad seeking to explore the social innovation story of Castel del Giudice and to understand the key drivers of its success.

In the case of Casa delle Agriculture Tullia e Gino association, the practice starts as deeply rooted in agriculture and food production, as the production relations and the connection with the territory are central questions in reconfiguring the community and building a local economy.

The proposal for a Food Policy (Local Food Plan) as an overarching driver of the different social and economic elements of the practice emerged for the first time during the discussion of the Green Night 2021. In this sense, territorial marketing began by revaluing abandoned resources such as land and minor traditional crops, especially in favour of local consumption. These activities, including the Green Night, are geared towards the local community and not towards the massive flow of tourists, mainly oriented towards the revamping of musical tradition crystallized in the events related to Pizzica music and dance.

Working on narrative and at the same time implementing concrete action has produced a new imagery of rural life providing a resignification of places that has left no room for compromise, while always seeking dialogue and confrontation.

A relevant difference with Castel del Giudice concerns the role of the municipality and other local authorities, which are in dialogue with the association without having a particularly relevant proactive role.

Casa delle AgriCulture activities have shown the local community how rural regeneration can be based on socially inclusive and environmentally sustainable actions, opening up new perspectives for those with different skills and interests who are motivated to remain in the area. The association fostered an increase in new entrants into agriculture, boosted the cultivation of traditional biotypes and the re-use of abandoned land, and raised awareness on the agroecological and multifunctional model.

In the case of Pontebernardo, the focus of regenerating actions is presently placed on a tourist offer: walkways with shepherds and their sheep during the summer and visits to the farms as a direct experience of this fundamental local activity. At the same time, local actors stress those aspects more strictly connected with production, biodiversity conservation (revitalization and maintenance of a specific sheep race, for example), and the small cheese and meat processing laboratory in the Centre of Territorial Interpretation shows again the central role of food and food choices as a pivotal element of the locality-building process as well as a consistent part of shared community actions. Moreover, shared rules about architecture recovery and transborder development of a pastoral/tourist route (La Routo) show awareness about a development process based on local culture and sense of place, anticipating regional/national laws addressing local policies toward local cultural heritage valorisation as a crucial element of territorial regeneration.

Concerning the rural development models we observed, in Pontebernardo the *excellent interaction* of local institutions and informal community groups with the universities and research centers in the territory according to a proactive *mix of endogenous and neo-endogenous rural processes* and a positive diffusion of *social innovation and multi-functionality* in agriculture and livestock farming invites critically reconsideration of top-down strategies of development in the local dimension. At the regional scale, another notable role has been evidently played by the regional legal framework, especially support for Ecomuseums; such *institutional support* has ensured and accompanied the action of the pastoral community on its way to safeguarding, improving, and revitalizing the local heritage and the cultural, social, and productive activities of the area coupled with *expert support* and cooperation with regional and extra regional universities and specialized centers of research. This, moreover, enhanced *multi-actor locally engagement and participation in rural public spaces* and the production of new narratives and representations in the local dimension [62]. In the specific case of the Ecomuseum of Pastoralism, the LAG seems to have cooperated in the definition and outlining of a *new way to think about the mountain and to define the roadmap for newcomers reinhabiting* as well as the "*restanza*" of young people in the Valley [55,63,64]. Similarly, cooperation with Maison de la Transhumance in the framework of the LAG's project "La Routo" has contributed to development of the idea of vagrant pastoralism and pathways as a powerful tourist attractor. Great value has been added to the creation and development of pastoral technical schools (such as the École des bergers in France and the ongoing process of creation of a National School of Pastoralism in Italy) as a form of knowledge transfer and continuity in intergenerational transmission of the practice, dissemination, and recovery of pastoral activities as alternative and suitable employment which can contribute to the repopulation of the area.

### 5. Conclusions

Several conclusions can be elaborated about the cases we have observed and commented on through the categories we fixed as the keys to enabling a critical interpretation and analysis of the reciprocal entanglements of sustainable development processes, biocultural heritage, sense of place, and belonginess. Ethnography, conducted both onsite and online, as well as insightful confrontations with other colleagues and students about this case, emerged along the EARTH Project Online Workshop dedicated to this experience and allowed us to advance final considerations about the specific entanglements between bio-cultural heritage, sense of place, and sustainable development based on the five items we chose to compare these three case studies.

*Biocultural heritage* is a powerful notion to observe across all three cases.

1.  In the case of Castel del Giudice, the reference to ancient memories and territorial vocations seems to be less intense than in the other cases. Regeneration in Castel del Giudice seems to pass more for social innovation and economic as well as socio-cultural innovation and inclusiveness. The uses of the past are above all linked to the memory of the ancient village where the innovative local action has been realized: the old emblematic and symbolic school restored to host old citizens and the ancient stables recuperated as a diffused hotel.

2.  In Castiglione d'Otranto, the references to ancient rural civilization and the knowledge–practice system are rooted in shared local memory, olive oil and traditional vegetable cultivation, the community mill based on ancient cultivated and evolutive grains, and the sociability promoted in the Green Night based on open spaces, sociability, fair-based encounters, and local traditional food and recipes.

3.  In Pontebernardo, the reference to biocultural heritage is particular intense based on natural recuperation (the resumption of the Sambucana sheep breeding) as a starting point for the local Consortium. Cultural revitalization through the specific research of the interpretation centre, recuperation of walkways and tools and ancient processing of raw materials such as sheep milk and meat to realize typical agri-food products are perfectly integrated in the framework of the ecomuseum's offerings and even in the actions proposed by the LAG concentrated on cross-border activities between Italian and French mountainous areas.

Recovered biodiversity, both cultivated and bred, represents both a crucial element of typical local products and first of all as a sediment of the socio-productive history of the place. This trend emerged early at the Ecomuseum of Pontebernardo and seems more recently to have spread its influence, moreso at Castel del Giudice than at Castiglione d'Otranto. This radically territorialized and *bio-cultural concern* makes the different communities addressed by the research more active, responsive, and performative in terms of local governance and markets as well as being at a regional scale, for example, via groups of responsible consumers.

Concerning *resilience,* the ethnographies with respect to the elements of old and new forms of rurality can be observed in all the cases, outlining a change in the exclusively endogenous regeneration processes: "affective entrepreneurship" in Castel del Giudice and the LAG's roles both in Pontebernardo and in Castiglione d'Otranto. The presence of outsiders has changed the local communities' perspectives about local resources, helping the communities to regain self-confidence and supporting local development, not only financially. These neo-endogenous actors enter the communities sharing and partially transforming their objectives, strategies, and development trajectories and activating different regeneration processes, all aimed at generating new opportunities for residents and creating conditions to attract them.

Regarding the *reduction of depopulation* through biocultural heritage valorization and rural innovative and inclusive practices, such as, for example, return and repopulation of the small urban/rural areas addressed by the paper, we had the opportunity to observe similar dynamics. The transformation and decline of family agriculture and extensive pastoralism seem the main cause of the demographic crisis observed in all the

cases proposed, which inexorably accelerated the further decline in a perverse spiral of impoverishment. The modernist productivism of the primary sector during the 1970s and 1980s certainly influenced the agricultural models underlying depopulation, the most serious economic and social consequences of which were suffered in precisely the more peripheral or mountainous areas to which the cases examined here belong. Access to public services and critical mobilities continue to represent the greatest difficulties in the addressed case studies. Such inadequacy of basic services could, however, still trigger a dangerous spiral of decline if the trend towards depopulation is not reversed in time; this can be greatly reduced by the interesting practices of social innovation implemented by enlightened local governance.

Meanwhile, on the front of *social innovation and peasant social inclusion*, all the cases underlie the same goal of community survival achieved through the activation, more or less consciously, of similar social innovation processes. In each of the examined cases, the process started with small groups of local actors who identified and proposed innovative solutions to meet the needs of their respective communities (stemming depopulation, increasing employment, regaining competitiveness) through interaction with actors and/or exogenous factors (returning entrepreneurs, cultural associations, LAGs, funding, management). The demonstration of the greater effectiveness of the new practices compared to those of the past has generated attention and consensus from other members of the community, who have joined in this innovation or borrowed it in a process of progressive involvement of the entire community, which has therefore modified practices and behaviors of each individual belonging to it, influencing the original development objectives and reorienting them towards neglected or unexplored areas of application.

Finally, regarding *participation and multi-actor involvement*, the above-described processes have empowered and motivated the communities, progressively stimulating the ability to rethink their own territory. The different experiences illustrated in each case are the result of new ways of social interaction, allowing the communities to progressively refine the tools in order to bring them closer to their respective development objectives, which have become ever more ambitious.

To ensure the success of the actions taken, the competence and creativity of the political actors did not appear secondary, especially in Castel del Giudice, where many actions were promoted and accompanied from below, triggering significant behaviors of normative isomorphism based on the mobilization of all local actors and re-generating solid links between the political and social components of the community. In Pontebernardo the socio-institutional interaction seems to be the real element of success of the regenerating action, and in Castiglione d'Otranto the pivotal element has to be considered a powerful and effervescent community of knowledge and practice composed of both local citizens and qualified newcomers attracted by the ongoing experimental and creative process.

Even though all these cases aim to address socio-cultural and economic regeneration in the respective localities, in Castel del Giudice innovation, new rural development models, and new forms of neo-endogenous influence in the regeneration process seem to prevail. In Pontebernardo, all the same items are represented in a very efficient and performative way, supported by the local community and local/regional institutions. In Castiglione d'Otranto, as in Castel del Giudice, the elements of fragility are probably represented by the less relevant influence of biocultural heritage and embeddedness of the local process of regeneration, which seems to correspond to lower levels of community commitment and participation. In the case of Castel del Giudice, this results in moderated involvement of citizens in the decision-making process, whereas Castiglione d'Otranto sees the participation of a powerful coordinating group which does not immediately correspond to wider local community participation.

Social innovation and inclusiveness seem to feed on symbolic rooting in the territories, on the memory of traditional agricultural practices, and on old forms of sociality though at different scales. Where this kind of embeddedness is realized, this must be considered as an added source of value of the regeneration actions undertaken and of the concrete and symbolic products deriving from recovered rural activities, resulting finally in even more impactful and strong participation.

**Author Contributions:** Conceptualization, L.B. and A.B.; methodology, L.B. and A.B.; investigation, L.B., A.B. and M.C.; writing—original draft preparation, L.B.; writing—review and editing, L.B., A.B. and M.C.; funding acquisition, L.B. All authors have read and agreed to the published version of the manuscript. The authors have also written different paragraphs: L.B. wrote specifically Ch. 1–Introduction up to page 4; Ch. 2–Materials and Methods; Ch. 3–Results–3.3. Pontebernardo; Ch. 4–Discussion (the part specifically dedicated to Pontebernardo) M.C. wrote specifically Results–3.2 Casa delle Agriculture; Ch. 4–Discussion (the part specifically dedicated to Casa delle Agri- Culture); the other parts were written by A.B.

**Funding:** This research was funded by EACEA–ERASMUS + PROGRAM EARTH Project, grant number 598839-EPP-1-2018-1-IT-EPPKA2-CBHE-JP and The APC was funded by University of Molise as Coordinator of the project.

**Institutional Review Board Statement:** We confirm that all subjects gave their informed consent for inclusion before they participated in the study. The study was conducted in accordance with the Declaration of Helsinki, and the protocol was approved by the Ethics Committee of EARTH + ERASMUS + Project (598839-EPP-1-2018-1-IT-EPPKA2-CBHE-JP).

**Informed Consent Statement:** Informed consent was obtained from all subjects involved in the study.

**Data Availability Statement:** All the interviews are available in a common author's drive which eventually can be shared on demand.

**Conflicts of Interest:** The authors declare no conflict of interest.

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
