# Peer review of "Sense of Place, Biocultural Heritage, and Sustainable Knowledge and Practices in Three Italian Rural Regeneration Processes"

_sustainability, doi:10.3390/su14084858_

Round 1

Reviewer 1 Report

sustainability-1662459, Knowledge and practices of sustainability: sense of places, biocultural heritage, and rural regeneration processes
Authors: Letizia Bindi , Mauro Conti , Angelo Belliggiano

Special Issue: Cultivating the Ecological Transition: Knowledge and Practices of Sustainability, Neo-Endogenous Development and Participatory Processes

koerner comments: Amongst many other thing, this extremely well researched article illustrates the very direct bearing that highly context sensitive (ethnographically grounded) analyses of cases studies of new forms of "rurality" have upon replacing 'top down' paradigm for 'sustainable development' with approaches that focus on the potential of 'bottom up' (endogenous) innovation in 'biocultural heritage' based rural 'resilience' and 'renovation'. The article's novel insights of the importance to 'sense of place' - and rural resilience of regenerating 'bio-cultural heritage' addresses critical gaps in current top down heritage conservation policy - that are likely to concern many contributions to the Special Issue. For instance - categories that tend to polarise 'natural' - 'cultural' heritage.  

Of special interest is the attention the article draws to the wider cross disciplinary significance of a renewed interest in socio-cultural anthropology in the topic of resilience in the study of "rural areas and peasant cultures" - see for example, especially Economic and Political Anthropology, e.g., Eric Wolf. I strongly recommend that the submission is accepted for publication - but wish to suggest a an area of improvement - as well as sources that might be useful for linking the aims to the significance of findings more directly.  

Area of improvement - please more reader friendly: these are surely very demanding topics - with implications for making dense writing necessary. But please edit to ensure clarity, reader accessibility and appreciation of novel connections between aims approaches and results.  

With regards to suggestions of sources to contextualise the approach:  

(1) the authors might look to the focus in Economic Anthropology and Political Anthropology of "rural areas and peasant cultures" - "resilience" - (especially 1960s - 1980s) as strategies for 'decentering' history - and, especially, what is meant by 'modernity' by focusing on "cases situated in different areas" of a country.  

(2) biocultural - the authors might look to approaches to 'biocultural' suggested by, for instance:  

Ingold, T. 2000 [1993]. The Perception of the Environment. Essays in livelihood, dwelling and skill. London: Routledge.  

Latour, B. and Weibel, P. (eds) 2021. Critical Zones. The Politics and Science of Landing on Earth. Cambridge, MA: MIT Press.  

(3) In Sustainability - on the importance of place for linking heritage conservation to sustainable development in contexts of deep conflict (a topic that relates directlyto 'resilience' - for instance:

Alsalloum, A., & Brown, A. (2019). Towards a Heritage-Led Sustainable Post-Conflict Reconciliation: A Policy-Led Perspective. Sustainability, 11(6). doi:10.3390/su11061686

Author Response

koerner comments: Amongst many other thing, this extremely well researched article illustrates the very direct bearing that highly context sensitive (ethnographically grounded) analyses of cases studies of new forms of "rurality" have upon replacing 'top down' paradigm for 'sustainable development' with approaches that focus on the potential of 'bottom up' (endogenous) innovation in 'biocultural heritage' based rural 'resilience' and 'renovation'. The article's novel insights of the importance to 'sense of place' - and rural resilience of regenerating 'bio-cultural heritage' addresses critical gaps in current top down heritage conservation policy - that are likely to concern many contributions to the Special Issue. For instance - categories that tend to polarise 'natural' - 'cultural' heritage.  

Of special interest is the attention the article draws to the wider cross disciplinary significance of a renewed interest in socio-cultural anthropology in the topic of resilience in the study of "rural areas and peasant cultures" - see for example, especially Economic and Political Anthropology, e.g., Eric Wolf. I strongly recommend that the submission is accepted for publication - but wish to suggest an area of improvement - as well as sources that might be useful for linking the aims to the significance of findings more directly.  

The a. agreed with the suggestions and critical observations and have added some bibliographical refences not only as items and quotes but also in the body of the text considering some notions have to be explained more attentively and implemented, exactly complying with the suggested attention to top-down heritage conservation policies, power issues and implications in the heritage conservation/valorisation process, a closer attention to “biocultural” notion as a contribution to the refinement and critical rethinking of the natural/cultural partage determined by some heritagization processes observed both in the general overview of the topics in the debate than in the fieldworks addressed.

Area of improvement - please more reader friendly: these are surely very demanding topics - with implications for making dense writing necessary. But please edit to ensure clarity, reader accessibility and appreciation of novel connections between aims approaches and results.  

The paper has been revised to comply with the request and suggestion to make it more easily readable and a bit more schematic in order to simplify or solve some theoretical passages and insights.

With regards to suggestions of sources to contextualise the approach:  

  • the authors might look to the focus in Economic Anthropology and Political Anthropology of "rural areas and peasant cultures" - "resilience" - (especially 1960s - 1980s) as strategies for 'decentering' history - and, especially, what is meant by 'modernity' by focusing on "cases situated in different areas" of a country.  

We inserted references and short discussion of the notions of rural regions and peasant cultures as well as of modernity referring to the political anthropological debates through the decades across the decades. See reference to Dalton, Wolf et al., Anderson, Firth, Geertz.

(2) biocultural - the authors might look to approaches to 'biocultural' suggested by, for instance:  

Ingold, T. 2000 [1993]. The Perception of the Environment. Essays in livelihood, dwelling and skill. London: Routledge.  

Latour, B. and Weibel, P. (eds) 2021. Critical ZonesThe Politics and Science of Landing on Earth. Cambridge, MA: MIT Press.  

We add some more bibliographical as well as definitions to the huge debate around biocultural heritage, summarizing both the latin-american reflection of “patrimonio biocultural” than the insightful reflection of Tim Ingold that is so close to our reflections that he has been generously prefacing even the last book of one of this paper’s authors specifically addressing pastoralism in the move as a biocultural heritage issue. We are, moreover, very glad of the last Latour’s book suggestion  which is summing up to other important book on environmental contradictions as political and cultural frictions.

(3) In Sustainability - on the importance of place for linking heritage conservation to sustainable development in contexts of deep conflict (a topic that relates directlyto 'resilience' - for instance:

Alsalloum, A., & Brown, A. (2019). Towards a Heritage-Led Sustainable Post-Conflict Reconciliation: A Policy-Led Perspective. Sustainability, 11(6). doi:10.3390/su11061686

The topic of socio-cultural mediation in the rural contexts and of negotiation and reconciliation of several conflicts and frictions inhabiting these small rural communities have been truly solicited by the lecture of the suggested paper that we gratefully added to the references of our paper.

Reviewer 2 Report

The article explores interesting research issues and has some potential. However, the material presented in this form has some significant weaknesses. In general, the aim of the work is blurred, there is no consistency between the main parts (title, aim, methodology, conclusions). There are many threads in the work, it is not known what the leading research issue is. As a result, it is not possible to evaluate some of the points required in the review and determines the low overall scientific value of the presented material.

The detailed comments are as follows:

  1. The title of the work is very generally worded, no indication that it is an example of Italy.
  2. There are no essential elements in the summary, i.e. a clearly stated goal and main conclusions from the research.
  3. The formulated goal is very vague, it does not refer to the elements of rusticity, significant in terms of the empirical part.
  4. In the introduction, some conclusions are not supported by any literature items, there are references to Italy - although it was not previously justified either in the title or in the text why it is important. For a better understanding of the presented issues, I would suggest changing the order of considerations, i.e. placing theoretical considerations before formulating the aim of the work.
  5. In the methodology, the description of the next research steps is vague - the question is when and how many interviews were carried out?
  6. The results presented only in descriptive form are relatively difficult to perceive - for a better reception of the research results, I suggest considering supplementing the description with some graphic elements.
  7. The discussion does not seem to take the classic form of discussion; no indication of the limitations of the study.
  8. In the fragments, the text is underdeveloped in the editing layer, e.g. fragments from the template appear in the references.

The article has potential, but it cannot be assessed positively in this form.

Author Response

The article explores interesting research issues and has some potential. However, the material presented in this form has some significant weaknesses. In general, the aim of the work is blurred, there is no consistency between the main parts (title, aim, methodology, conclusions). There are many threads in the work, it is not known what the leading research issue is. As a result, it is not possible to evaluate some of the points required in the review and determines the low overall scientific value of the presented material.

The detailed comments are as follows:

  1. The title of the work is very generally worded, no indication that it is an example of Italy.

We specified and refine the title of the paper

  1. There are no essential elements in the summary, i.e. a clearly stated goal and main conclusions from the research.

We provided a more schematic summary,

  1. The formulated goal is very vague, it does not refer to the elements of rusticity, significant in terms of the empirical part.

We describe goals adding points and hopefully clearer outputs of the research, trying to make more strict connection with the preliminary reflections

  1. In the introduction, some conclusions are not supported by any literature items, there are references to Italy - although it was not previously justified either in the title or in the text why it is important. For a better understanding of the presented issues, I would suggest changing the order of considerations, i.e. placing theoretical considerations before formulating the aim of the work.

We move some parts to other sections of the paper, but above all we added some sections to the paper in order to explain the relevance both in terms of comparison than of specific ethnographic insights to address the three Italian cases we choose and why the present phase of Italian post-covid recovery and resilience plan is adding value to such a research.

  1. In the methodology, the description of the next research steps is vague - the question is when and how many interviews were carried out?

We precisely clarify this point in the section ‘Materials’ of the paper. We added moreover a more schematic lists of the overall methodological strategy and tools / techniques used during interviews, focus gropus, interactive meetings and so on in order to outline in a more precise form all the moments and choices of the research.

  1. The results presented only in descriptive form are relatively difficult to perceive - for a better reception of the research results, I suggest considering supplementing the description with some graphic elements.
  2. The discussion does not seem to take the classic form of discussion; no indication of the limitations of the study. 
  3. In the fragments, the text is underdeveloped in the editing layer, e.g. fragments from the template appear in the references.

This oversight has been corrected

Reviewer 3 Report

I would recommend improving the focus of the paper. Too wordy text, the focus is blurred, it is rather hard to capture the main ideas and arguments and follow them. The author should clearly define the aim, structure the gaps the paper aims to address, structure key findings (point by point), and discuss these findings. In the conclusion, the author should clearly focus on demonstrating whether the aim has been achieved, what are the results (points), how novel they are, what are the implications, limitations, etc. The methodology must be detailed. It is not clear how the cases were selected (approach should be detailed), how they were interviewed (questionnaire or what?), how the interview results were unified and measured.   

Author Response

I would recommend improving the focus of the paper.

The focus, methodology, definition of the cases and materials/data/ documents used for the research have been refined and attentively described and inserted.

Too wordy text, the focus is blurred, it is rather hard to capture the main ideas and arguments and follow them.

We reduce words, adjectives, some phrases have been refined, shortened, inserted more punctuation, more paratactic structure and more points to describe specific and relevant elements of the discussion. We hope this made the paper more readable and fluent.

The author should clearly define the aim, structure the gaps the paper aims to address, structure key findings (point by point), and discuss these findings.

This has been added both to the methodological section than to the discussion part of the paper.

In the conclusion, the author should clearly focus on demonstrating whether the aim has been achieved, what are the results (points), how novel they are, what are the implications, limitations, etc.

The methodology must be detailed. It is not clear how the cases were selected (approach should be detailed), how they were interviewed (questionnaire or what?), how the interview results were unified and measured.   

This part has been added and refined.

Round 2

Reviewer 2 Report

  1. The article needs to be supplemented with regard to research limitations. What limitations do the Authors see in their research, especially taking into account the fact that there were different ways of conducting interviews (directly and using the Internet)?
  2. There will be abbreviations in the work that are not preceded by an appropriate explanation, e.g. LAG abbreviation in line 192 and explanation in line 542
  3. The article still requires editing corrections, there are many text highlights, etc.
  4. Line 69, 1018 - what century did the Authors mean?

Author Response

  1. The article needs to be supplemented with regard to research limitations. What limitations do the Authors see in their research, especially taking into account the fact that there were different ways of conducting interviews (directly and using the Internet)?

327-330 “However, the use of different survey criteria, imposed by the particular pandemic conditions, did not imply significant problems of involvement and / or participation of the actors, nor of detection and interpretation of the data, thanks to the author’s long-term and deep awareness of local contexts [30-31; 51].

  1. There will be abbreviations in the work that are not preceded by an appropriate explanation, e.g., LAG abbreviation in line 192 and explanation in line 542

191 (LAG Abbreviation is solved at this line) + 542

  1. The article still requires editing corrections, there are many text highlights, etc.

We didn’t notice particular editing mistakes except at pages: 1024-1051 which was in bold; at pag. 1001 interline spaces and other small mistakes which have been corrected.

  1. Line 69, 1018 - what century did the Authors mean?

69: XX Century

Reviewer 3 Report

In general, my recommendations have been addressed. However, I still consider the text too unfocused and excessively long. For instance, the three-page Conclusion. I highly recommend the authors trim the text and clearly emphasize major findings (probably, a bulleted list, focused and brief discussion of findings one by one, focus on novelty).

Author Response

In general, my recommendations have been addressed. However, I still consider the text too unfocused and excessively long. For instance, the three-page Conclusion. I highly recommend the authors trim the text and clearly emphasize major findings (probably, a bulleted list, focused and brief discussion of findings one by one, focus on novelty).

See, please, the moderate reformulation of the conclusions both in terms of reduction of words than in term of emphasis and systematization of data in the conclusive part of the paper.